# A Soft–Hard Combination Decision Fusion Scheme for a Clustered Distributed Detection System with Multiple Sensors

**DOI:** 10.3390/s18124370

**Published:** 2018-12-10

**Authors:** Junhai Luo, Xiaoting He

**Affiliations:** School of Information and Communication Engineering, University of Electronic Science and Technology of China, Chengdu 611731, China; 201722021138@std.uestc.edu.cn

**Keywords:** distributed detection system, soft decision, hard decision, clustering, fuzzy logic system, fuzzy c-means algorithm

## Abstract

In the distributed detection system with multiple sensors, there are two ways for local sensors to deliver their local decisions to the fusion center (FC): a one-bit hard decision and a multiple-bit soft decision. Compared with the soft decision, the hard decision has worse detection performance due to the loss of sensing information but has the main advantage of smaller communication costs. To get a tradeoff between communication costs and detection performance, we propose a soft–hard combination decision fusion scheme for the clustered distributed detection system with multiple sensors and non-ideal communication channels. A clustered distributed detection system is configured by a fuzzy logic system and a fuzzy c-means clustering algorithm. In clusters, each local sensor transmits its local multiple-bit soft decision to its corresponding cluster head (CH) under the non-ideal channel, in which a simple and efficient soft decision fusion method is used. Between clusters, the fusion center combines all cluster heads’ one-bit hard decisions into a final global decision by using an optimal fusion rule. We show that the clustered distributed system with the proposed scheme has a good performance that is close to that of the centralized system, but it consumes much less energy than the centralized system at the same time. In addition, the system with the proposed scheme significantly outperforms the conventional distributed detection system that only uses a hard decision fusion. Using simulation results, we also show that the detection performance increases when more bits are delivered in the soft decision in the distributed detection system.

## 1. Introduction

Multiple-sensor data fusion has attracted significant attention in the information fusion field. It can be mainly divided into two types, namely centralized data fusion and distributed data fusion. In a centralized data fusion system, sensing information (raw sensing data) observed by local sensors is delivered directly to the fusion center (FC) through single hop or multiple hops. The FC collects all the data while carrying out the computations and makes a final decision. A centralized data fusion system can get optimal detection performance due to the small loss of information. However, it is at the cost of a large bandwidth and a heavy computing burden of the FC, which increases the system’s communication costs and shortens the system’s lifetime. Communication costs mainly include the cost for the sensors used in the sensing system, the communication bandwidth required by the system, which determines the number of bits that could be transferred by sensors, and all the energy consumption of data transmission. Compared with centralized data fusion, distributed data fusion has been studied more for several decades because it bears the advantages of higher reliability, smaller communication costs, higher survivability, and shorter decision time than centralized data fusion. In a distributed fusion system, each local sensor makes a local decision based on its own observation and transmits it to the FC. Then, the FC combines all local decisions into a final global decision by using an optimal decision rule or a suboptimal decision rule [1,2].

There are three basic fusion topology architectures in the distributed detection system, namely parallel topology [3,4], serial topology [5,6,7], and tree topology [8]. For the serial topology and tree topology, the main disadvantage of them is that any local sensor’s failure would result in the whole system’s failure. However, for parallel topology, one or several local sensors’ failure would not affect other local sensors because that each local sensor works independently. Therefore, we adopt the distributed detection system with the structure of parallel topology, whose typical structure is shown in Figure 1, in this paper. 

In the distributed detection system with multiple sensors, there are two ways for local sensors to deliver their local decisions to the FC: a one-bit hard decision and a multiple-bit soft decision. For a hard decision, only one bit is transmitted to the FC. Usually, Bit 0 represents the absence of the target and Bit 1 represents the presence of the target. It needs less bandwidth and reduces the system’s communication costs but loses a lot of sensing data information and restricts the improvement of the system’s detection performance [9,10,11]. In fact, a multiple-bit soft decision can be transmitted within the system’s communication capability. A Bayesian model has been proposed in Reference [12], in which each local sensor delivers a probability that represents its confidence in its decision to the FC. Then, the FC combines all the probabilities into a global decision. Thomopoulos et al. proposed a soft decision scheme in which a two-bit soft information was delivered by each local sensor to the FC [13]. The two-bit soft information not only contains one-bit hard decision information regarding the presence of the target but also contains one-bit of quality information. In Reference [14], Lee and Chao proposed a multiple-bit soft fusion scheme based on subpartitioning of the local decision space. In Reference [15], Aziz proposed a multilevel quantization and fusion approach based on fuzzy techniques. However, most of those proposed methods are computationally complex, especially with multilevel quantization, and they did not take the non-ideal communication channel into consideration. In Reference [16], the author optimized the number of reporting bits to maximize the network’s throughput in quantized cooperative spectrum sensing. In Reference [17], the number of reporting bits and the combining weight were jointly optimized to maximize the probability of detection. Therefore, inspired by References [14,15], we propose a simple, easy to calculate, and efficient soft decision fusion method based on the subpartitioning of the local decision space and considers the non-ideal communication channel. In addition, to get a tradeoff between communication costs and detection performance, we also propose a soft–hard combination decision fusion scheme (SH-DFS) for the distributed detection system.

Clustering techniques have been widely studied and used in the distributed detection system [18,19,20,21,22,23]. They can not only reduce the system’s energy consumption but also extend local sensors’ and the system’s lifetime. In the clustered distributed detection system, every cluster has a cluster head (CH) and each local sensor belongs to a cluster. First, each local sensor delivers its decision to its corresponding CH. Then, each CH makes its own decisions according to the received data and delivers it to the FC. Finally, the FC fuses the received information and makes a final decision. There are numerous methods for CHs selection and clusters formation proposed by previous studies. In Reference [24], the authors proposed an easy method for CHs selection, in which each local sensor has an equal probability to be CH. In the cluster head election mechanism using fuzzy logic (CHEF), CHs are selected based on two parameters, which are proximity distance and energy [25]. In Reference [26], the energy efficient structured clustering algorithm (EESCA) is proposed, in which CH is elected based on average communication distance and lingering energy. In Reference [27], the author proposed adaptive dynamic clustering (ADC) to minimize the cluster head and improve the network’s routing problem. In this paper, we select CHs by using a fuzzy logic system (FLS), in which the remaining energy and distance to the FC are considered to compute the likelihood to be a CH for each local sensor. After selecting CHs, clusters are formed by using the fuzzy c-means clustering algorithm because that the fuzzy c-means clustering algorithm has a more accurate and natural description of data [11].

In this paper, we propose a soft–hard combination decision fusion scheme for the clustered distributed detection system with multiple sensors. First, an FLS is designed to select a CH, in which the remaining energy and distance to the FC are considered to compute the likelihood to be a CH for each local sensor. Then, the fuzzy c-means clustering algorithm is used for forming clusters. In every cluster, soft decision fusion based on the subpartitioning of the local decision space is applied. Every local sensor delivers its multiple-bit decision, which not only contains its decision but also contains its degree of confidence on that decision to its corresponding CH. CH combines all received data into a one-bit hard decision under the Neyman–Pearson criterion. Between clusters, the FC combines all cluster heads’ one-bit hard decisions into a final global decision by using an optimal fusion rule. In comparison with previous approaches, the novelty and contributions of this paper are summarized as follows:We propose a soft–hard combination decision fusion scheme, which not only makes use of soft decision fusion but also makes uses of hard decision fusion. This scheme can get a tradeoff between communication costs and detection performance.In every cluster with soft decision fusion, a simple, easy to calculate, and efficient soft decision fusion method based on the subpartitioning of the local decision space is applied. Multiple bits that not only contain a local sensor’s decision but also contain its degree of confidence on that decision are delivered. At the same time, compared with References [14,15], our soft decision fusion method simplifies the calculation of CHs.The non-ideal communication channel is taken into consideration.In the FC, an optimal fusion rule is applied to fuse all decisions form all clusters.

The paper is organized as follows. Section 2 shows the design scheme of the clustered distributed detection system. Then, the soft decision fusion scheme in clusters based on the subpartitioning of the local decision space is presented in Section 3. Section 4 shows the hard fusion scheme with an optimal fusion rule. Simulation results and performance evaluations are presented in Section 5. Section 6 concludes this paper.

## 2. The Design of Clustered Detection System

We consider a distributed detection system consisting of N local sensors. We design the FLS to select CHs and use the fuzzy c-means clustering algorithm for clusters formation. In addition, the clustered distributed detection system can be dynamically reconfigured in every round.

### 2.1. The Design of Fuzzy Logic System

In the distributed detection system, CHs not only need to fuse multiple-bit soft information in clusters but also need to send one-bit hard information to the FC. Therefore, CHs must have sufficient energy and be close to the FC. Therefore, two antecedents, including every local sensor’s remaining energy and distance to the FC, are considered to design the FLS. Therefore, two antecedents of a local sensor are considered in our designed FLS:Antecedent 1: every local sensor’s remaining energy.Antecedent 2: every local sensor’s distance to the FC.

The linguistic variables used to represent antecedent 1 are divided into three levels: low, moderate, and high; and those to represent antecedent 2 are also divided into three levels: near, moderate, and far. Two antecedents use the same kind of membership functions (MFs), and in this paper, MFs for normalized antecedents are shown in Figure 2. Two antecedents as the input of MFs have been normalized to [0, 10]. The linguistic variables, which are used to represent the consequent and denote the possibility that a local sensor will be selected as a CH, were divided into five levels: very small, small, medium, large, and very large. In this paper, MFs for normalized consequents are shown in Figure 3. 

Based on the fact that an ideal CH should have sufficient energies and be close to the FC, we design our FLS using rules for every input (*x*_1_, *x*_2_) like:

R^l^: IF the remaining energies of a local sensor (*x*_1_) is $F_{1}^{l}$, and its distance to the FC (*x*_2_) is $F_{2}^{l}$, THEN the possibility that this local sensor will be selected as a CH(*y*) is G^l^, where l = 1, …, 9. Nine rules are summarized in Table 1.

For every input (*x*_1_, *x*_2_) of each local sensor, the output is computed using:(1)$$Y\left(x_{1},\;x_{2}\right)=\frac{\sum_{l=1}^{9}\mu_{F_{1}^{l\;}\left(x_{1}\right)\;}\mu_{F_{2}^{l\;}\left(x_{2}\right)}c^{l}}{\sum_{l=1}^{9}\mu_{F_{1}^{l\;}\left(x_{1}\right)\;}\mu_{F_{2}^{l\;}\left(x_{2}\right)}}\;$$

According to the output, which is the probability of a local sensor selected as a CH, we can select M local sensors with the highest probabilities as CHs. M is the number of CHs, and it can be obtained using:(2)M=floor(p×N)
where p is a constant ratio decided by people and represents the proportion of CHs in all sensors, and N is the total number of local sensors in the distributed detection system.

### 2.2. The Fuzzy c-Means Clustering Algorithm

After selecting CHs by using FLS, the fuzzy c-means algorithm for clustering is used for forming clusters. Compared with hard clustering, in which every local sensor only belongs to one cluster, FCM has a more accurate and natural description of data. Let {*x_i_*, *i* = 1, 2, …, N} be the set of N local sensors, and *x_i_* represents the *i*th local sensor. M is the number of CHs computed by Equation (2) and is also the number of clusters in the system. 

The fuzzy c-means algorithm is an improvement of the c-means algorithm [28]. The c-means algorithm is based on minimizing the following objective function, which represents the mean-square error: (3)$$\min J_{e}=\overset{M}{\underset{i=1}{\sum }}\underset{y\in C_{j}}{\sum }||y-m_{i}||{}^{2}$$
where *C_j_* represents the cluster of *j*, ***m****_i_* is the center of *C_i_*, and ***y***(***y***∈*C_j_*) represents all local sensors in *C_j_*. Different from the c-means algorithm, the fuzzy c-means algorithm is based on minimizing the following objective function under one restriction function:(4)$$\{\begin{array}{c} \min J_{f}=\sum_{j=1}^{M}\sum_{i=1}^{N}{\left[\mu_{j}\left(x_{i}\right)\right]}^{b}||x_{i}-m_{j}||{}^{2}\\ \mathrm{st}.\;\sum_{j=1}^{M}\mu_{j}\left(x_{i}\right)=1,\;\;i=1,\;2,\;\ldots ,\;N \end{array}$$
where ***m_j_*** is the center of cluster *j*, $\mu_{j}\left(x_{i}\right)$ is the membership of the *i*th local sensor in cluster *j*, and b is a constant that can control the degree of blurring of clustering results. If *b*→1, clusters formed by the fuzzy c-means algorithm is similar to those formed by c-means. If *b* = ∞, the fuzzy c-means algorithm will get a completely fuzzy solution, which means every node belongs to each cluster with equal probability, but it does not bear the meaning of clustering. Generally, let *b* = 2. In addition, the fuzzy c-means algorithm requires that the sum of the memberships of one node for each cluster is 1, which is illustrated by Equation (4).

The solution of Equation (4) can be obtained by optimizing $\mu_{j}\left(x_{i}\right)$ and update ***m****_j_* according to Equation (5) through an iterative method:(5)$$\{\begin{array}{ccc} \mu_{j}\left(x_{i}\right) & = & \frac{{\left(1/||x_{i}-m_{j}||{}^{2}\right)}^{1/\left(b-1\right)}}{\sum_{k=1}^{M}{\left(1/||x_{i}-m_{k}||{}^{2}\right)}^{1/\left(b-1\right)}}\\ m_{j} & = & \frac{\sum_{i=1}^{n}{\left[\mu_{j}\left(x_{i}\right)\right]}^{b}x_{i}}{\sum_{i=1}^{n}{\left[\mu_{j}\left(x_{i}\right)\right]}^{b}} \end{array}$$

The specific description of clusters formed by FCM is illustrated in Algorithm 1.

**Algorithm 1** Clusters formation by FCM**Input:**   *x*_i_, *i* = 1, 2, …, *n*; M; *b*; *μ_j_*(***x_i_***); max_iteration_num      M: the number of clusters and obtained by Equation (2).      *b*: *b* = 2.       *μ_j_*(***x_i_***), *j* = 1, 2…, M, and *i* = 1, 2…, N: the initialized membership values.      max_iteration_num: the maximum number of iterations.**Output**: $\mu_{j}^{k}\left(x_{i}\right)$, which is the membership of each node belonging to each cluster for the *k*th iteration.
**Iteration Process:**
  $while\left\{\left(\mu_{j}^{k}\left(x_{i}\right)-\mu_{j}^{k-1}\left(x_{i}\right)\right)>e\;\;and\;\;k<\max\_\mathrm{iteration}\_\mathrm{num}\right\}\;\;\mathrm{do}$  $m_{j}=\frac{\sum_{i=1}^{n}{\left[\mu_{j}\left(x_{i}\right)\right]}^{b}x_{i}}{\sum_{i=1}^{n}{\left[\mu_{j}\left(x_{i}\right)\right]}^{b}}\;\;\;\;\;\;\;\;\;\forall \;j=1,\;2,\;\ldots ,\;M$  $\mu_{j}\left(x_{i}\right)=\frac{{\left(1/||x_{i}-m_{j}||{}^{2}\right)}^{1/\left(b-1\right)}}{\sum_{k=1}^{M}{\left(1/||x_{i}-m_{k}||{}^{2}\right)}^{1/\left(b-1\right)}}\;\;\;\;\;\;\forall \;i=1,\;2,\;\ldots ,\;n;\;j=1,\;2,\;\ldots ,\;M$
end while



## 3. Soft Decision Fusion Based on the Subpartitioning of the Local Decision Space in Clusters

We consider the binary detection in the clustered distributed detection system. In binary detection, there are two hypotheses: *H*_0_ represents the absence of the target (the signal) and *H*_1_ represents the presence of the target (the signal). The FC makes a global binary decision (0 or 1) by processing local decisions received from all local sensors. It is the hard decision fusion when each local sensor makes a binary hard decision (0 or 1) and delivers it to the FC. Conversely, it is the soft decision fusion when each local sensor delivers a multiple-bit decision to the FC. In this paper, we apply soft decision fusion in clusters formed by Section 2 and apply hard decision fusion between clusters. The proposed clustered detection system is shown in Figure 4.

We assume that there are N local sensors in the distributed detection system. By using the FLS and the fuzzy c-means clustering algorithm in Section 2, M clusters are formed. In the *m*th cluster, there are *n_m_* local sensors and *c_m_* is the CH in it. N = *n*_1_ + *n*_2_ + … + *n_M_*. We take the *m*th cluster for example.

In the *m*th cluster, let *y_1_*, *y_2_*, …, *y_nm_* be the statistically independent observations of *n_m_* local sensors and have known probability distributions under both hypotheses (*H*_0_ and *H*_1_). For the *k*th local sensor in the *m*th cluster, *T_k_* is the sensor’s threshold determined by the probability of false alarm of the *k*th local sensor, and *L_k_*(*y_k_*) is the likelihood ratio test at the *k*th local sensor, which is given by:$$L_{k}\left(y_{k}\right)=\frac{P\left(y_{k}|H_{1}\right)}{P\left(y_{k}|H_{0}\right)}$$

If a local decision *u_k_* is a one-bit hard decision, then the associate decision space Ω^k^ is partitioned into two exclusive regions, $\Omega_{0}^{k}$ and $\Omega_{1}^{k}$, such that:(6)$$\{\begin{array}{c} u_{k}=1,\;\;\;\;\;if\;\;y_{k}\in \Omega_{1}^{k}\;\;\\ u_{k}=0,\;\;\;\;if\;\;y_{k}\in \Omega_{0}^{k}\; \end{array}$$

Furthermore, we have a probability of detection (*P_dk_*) and a probability of false alarm (*P_fk_*) for the *k*th local sensor:(7)$$\{\begin{array}{c} P_{dk}=P\left\{u_{i}=1|H_{1}\right\}.\\ P_{fk}=P\left\{u_{i}=1|H_{0}\right\}. \end{array}$$

To improve the detection performance, a *b*-bit (*b* ≥ 2) soft decision can be obtained, in which *b* is determined by the communication capability of the system. Its essence is the subpartitioning of the $\Omega_{0}^{k}$ and $\;\Omega_{1}^{k}$ space. For illustration purposes, the 2-bit soft decision case is considered.

In the 2-bit soft decision case, $\Omega_{0}^{k}$ and $\Omega_{1}^{k}$ are partitioned into two exclusive regions respectively. 

The quantization rule and decision rule
(8)$$u_{k}=\{\begin{array}{l} 00\left(decision\;is\;H_{0}\;with\;higher\;confidence,\;soft\;decision\;value:0\right),\;\;\;\;\;y_{k}\in \Omega_{00}^{k}\\ 01\left(decision\;is\;H_{0}\;with\;lower\;confidence,\;soft\;decision\;value:1\right),\;\;\;\;\;y_{k}\in \Omega_{01}^{k}\\ 10\left(decision\;is\;H_{1}\;with\;lower\;confidence,\;soft\;decision\;value:2\right),\;\;\;\;\;y_{k}\in \Omega_{10}^{k}\\ 11\left(decision\;is\;H_{1}\;with\;higher\;confidence,\;soft\;decision\;value:3\right),\;\;\;\;\;y_{k}\in \Omega_{11}^{k} \end{array}$$
where:(9)$$\{\begin{array}{l} \Omega^{k}=\Omega_{0}^{k}\cup \Omega_{1}^{k},\;\;\Omega_{0}^{k}\cap \Omega_{1}^{k}=\emptyset \\ \Omega_{0}^{k}=\Omega_{00}^{k}\cup \Omega_{01}^{k},\;{\;\Omega }_{00}^{k}\cap \Omega_{01}^{k}=\emptyset \;\\ \Omega_{1}^{k}=\Omega_{10}^{k}\cup \Omega_{11}^{k},\;{\;\Omega }_{10}^{k}\cap \Omega_{11}^{k}=\emptyset \; \end{array}$$

The subpartitioning in the 2-bit decision case is shown in Figure 5.

Corresponding to Equation (7), we also have:(10)$$\{\begin{array}{c} \begin{array}{c} P_{d10k}=P\left\{u_{k}=10|H_{1}\right\}\\ P_{d11k}=P\left\{u_{k}=11|H_{1}\right\} \end{array}\\ \begin{array}{c} P_{f10k}=P\left\{u_{k}=10|H_{0}\right\}\\ P_{f11k}=P\left\{u_{k}=11|H_{0}\right\} \end{array} \end{array}$$

To simplify the calculation, we let the subpartition satisfy Equation (11) according to the probability of detection and the probability of false alarm:(11)$$\{\begin{array}{c} P_{d10k}=P_{d11k}=\frac{P_{dk}}{2}\;\\ P_{f10k}=P_{f11k}=\frac{P_{fk}}{2}\; \end{array}$$

According to the knowledge of the *k*th local sensor’s probability of false alarm and the signal-to-noise ratio (SNR), which is defined as the ratio of signal power to the noise power, we can get *P_dk_* and the range of sub-spaces.

Let t_0_ be the threshold between $\Omega_{00}^{k}$ and $\Omega_{01}^{k}$, t_1_ be the threshold between $\;\Omega_{01}^{k}$ and $\;\Omega_{10}^{k}$, and *t_2_* be the threshold between $\;\Omega_{10}^{k}$ and $\;\Omega_{11}^{k}$, then the calculation of the *k*th local sensor’s thresholds (*t_0_*, *t_1_*, *t_2_*) in the 2-bit soft decision case is summarized using two steps. Step 1: Given the false alarm probability of the *k*th local sensor *P_fk_*, then t_1_, which is not only the threshold between $\Omega_{01}^{k}$ and $\Omega_{10}^{k}$ but also the threshold between $\Omega_{0}^{k}$ and $\Omega_{1}^{k}$, can be obtained under the Neyman–Pearson criterion. *P_dk_* can be obtained using t_1_.

Step 2: Then, we can calculate t_0_ by making *P_f_*_10*k*_ = *P_f_*_11*k*_ = *P_fk_*/2, and calculate *t_2_* by making *P_d_*_10*k*_ = *P_d_*_11*k*_ = *P_dk_*/2.

Considering that the communication channel is non-ideal without memory, we assume that the bit error rate of the *k*th local sensor is *P_ek_* , which is the same for every bit. Then, for the *k*th local sensor, the probability of the bit being transmitted correctly is 1 − *P_ek_*. In the 2-bit decision case, the station transition diagram is shown in Figure 6. We can obtain that the probability of a 2-bit soft decision being transmitted correctly by the *k*th local sensor is (1 − *P_ek_*)^2^, the probability of one bit being transmitted wrongly is *P_ek_*(1 − *P_ek_*), and the probability of two bits all being transmitted wrongly is (*P_ek_*)^2^. Then, we can get the possible transition matrix of Figure 6:(12)$$PT_{k}=[\begin{array}{c} {\left(1-P_{ek}\right)}^{2}\\ \left(1-P_{ek}\right)P_{ek}\\ \left(1-P_{ek}\right)P_{ek}\\ {\left(P_{ek}\right)}^{2} \end{array}\begin{array}{c} \left(1-P_{ek}\right)P_{ek}\\ {\left(1-P_{ek}\right)}^{2}\\ {\left(P_{ek}\right)}^{2}\\ \left(1-P_{ek}\right)P_{ek} \end{array}\begin{array}{c} \left(1-P_{ek}\right)P_{ek}\\ {\left(P_{ek}\right)}^{2}\\ {\left(1-P_{ek}\right)}^{2}\\ \left(1-P_{ek}\right)P_{ek} \end{array}\begin{array}{c} {\left(P_{ek}\right)}^{2}\\ \left(1-P_{ek}\right)P_{ek}\\ \left(1-P_{ek}\right)P_{ek}\\ {\left(1-P_{ek}\right)}^{2} \end{array}]$$

Let ***U_m_*** = (*u*_1_, *u*_2_, …, *u_nm_*) be the decision vector of all local sensors in the *m*th cluster. The optimal decision rule for the CH in the *m*th cluster (*c_m_*) is as follows:(13)$$L\left(U_{m}\right)=\frac{P\left(U_{m}|H_{1}\right)}{P\left(U_{m}|H_{0}\right)}\{\begin{array}{c} >T_{m\;\;\;\;\;}\;\;decision\;is\;H_{1}\\ <T_{m\;\;\;\;\;}\;\;decision\;is\;H_{0}\\ =T_{m\;\;\;\;\;}decision\;is\;H_{1\;}with\;Pr\;\lambda \;\mathrm{and}\;\mathrm{decision}\;\mathrm{is}\;\;H_{0}\;with\;Pr\;\left(1-\lambda \right)\; \end{array}$$
where *T_m_* is the *m*th cluster’s threshold and is determined by the probability of the CH’s false alarm and λ is a randomization parameter. 

Assuming independence between local sensors, the likelihood ratio in the *m*th cluster is given using:(14)$$L\left(U_{m}\right)=\frac{P\left(U_{m}|H_{1}\right)}{P\left(U_{m}|H_{0}\right)}=\frac{P\left(u_{1},\;u_{2},\ldots ,u_{n_{m}}|H_{1}\right)}{P\left(u_{1},\;u_{2},\ldots ,u_{n_{m}}|H_{0}\right)}=\overset{n_{m}}{\underset{k=1}{\prod }}\frac{P\left(u_{k}|H_{1}\right)PT_{k}}{P\left(u_{k}|H_{0}\right)PT_{k}}$$
where:$$\{\begin{array}{l} P_{d10k}=P\left\{u_{k}=10|H_{1}\right\}=P_{d11k}=P\left\{u_{k}=11|H_{1}\right\}=\frac{P_{dk}}{2}\\ P_{m10k}=P\left\{u_{k}=00|H_{1}\right\}=P_{m01k}=P\left\{u_{k}=01|H_{1}\right\}=\frac{1-P_{dk}}{2}\\ P_{f10k}=P\left\{u_{k}=10|H_{0}\right\}=P_{f11k}=P\left\{u_{k}=11|H_{0}\right\}=\frac{P_{fk}}{2}\\ P_{10k}=P\left\{u_{k}=00|H_{0}\right\}=P_{11k}=P\left\{u_{k}=01|H_{0}\right\}=\frac{1-P_{fk}}{2}\;\;\;\;\; \end{array}$$

Therefore, the probability of a false alarm and the probability of detection of the CH can be obtained using:(15)$$P_{fc_{m}}=\underset{L\left(U_{m}\right)>T_{m\;\;\;\;\;}}{\sum }P\left(U_{m}|H_{0}\right)PT+\;\lambda \underset{L\left(U_{m}\right)=T_{m\;\;\;\;\;}}{\sum }P\left(U_{m}|H_{0}\right)$$
and:(16)$$P_{dc_{m}}=\underset{L\left(U_{m}\right)>T_{m\;\;\;\;\;}}{\sum }P\left(U_{m}|H_{1}\right)+\;\lambda \underset{L\left(U_{m}\right)=T_{m\;\;\;\;\;}}{\sum }P\left(U_{m}|H_{1}\right)$$

According to the probability of a false alarm of local sensor *c_m_* (CH), we can obtain *T_m_* and λ. Then, the probability of detection of the CH can be obtained using Equation (16).

To get a binary decision in the *m*th cluster when knowing *T_m_* and λ after calculation, we can follow:(17)$$\{\begin{array}{c} u_{c_{m}}=1,\;\;\;\;\;if\;\;L\left(U_{m}\right)>T_{m\;}\;or\;\;\lambda \geq 0.5\;when\;\;L\left(U_{m}\right)=T_{m\;}\;\\ u_{c_{m}}=0,\;\;\;\;\;if\;\;L\left(U_{m}\right)<T_{m\;}\;or\;\;\lambda <0.5\;when\;\;L\left(U_{m}\right)=T_{m\;} \end{array}$$

In the same way, for a *b*-bit soft decision, *b_n_* = 2^(*b*^^−1)^ − 1, the quantization rule and decision rule:(18)$$u_{k}=\{\begin{array}{cc} Binary\left(0\right)\left(decision\;is\;H_{0}\;with\;higher\;confidence\right), & y_{k}\in \Omega_{00}^{k}\\ Binary\left(1\right)\left(decision\;is\;H_{0}\;with\;confidence\;less\;than\;00\right), & y_{k}\in \Omega_{01}^{k}\\ \cdots & \\ Binary\left(2^{b-1}-1\right)\left(decision\;is\;H_{0}\;with\;the\;lowerwst\;confidence\right), & y_{k}\in \Omega_{0b_{n}}^{k}\\ Binary\left(2^{b-1}\right)\left(decision\;is\;H_{1}\;with\;the\;lowest\;confidence\right), & y_{k}\in \Omega_{10}^{k}\\ Binary\left(2^{b-1}+1\right)\left(decision\;is\;H_{1}\;with\;higher\;confidence\;than\;10\right), & y_{k}\in \Omega_{11}^{k}\\ \ldots & \\ Binary\left(2^{b}-1\right)\left(decision\;is\;H_{1}\;with\;the\;highest\;confidence\right), & y_{k}\in \Omega_{1b_{n}}^{k} \end{array}$$
where *Binary*() is the data in binary form and: (19)$$\{\begin{array}{cc} \Omega^{k}=\Omega_{0}^{k}\cup \Omega_{1}^{k}, & {\;\Omega }_{0}^{k}\cap \Omega_{1}^{k}\cap \ldots \cap \Omega_{bn}^{k}=\emptyset \;.\\ \Omega_{0}^{k}=\Omega_{00}^{k}\cup \Omega_{01}^{k}\cup \ldots \cup \Omega_{0b_{n}}^{k}, & {\;\Omega }_{00}^{k}\cap \Omega_{01}^{k}\cap \ldots \cap \Omega_{0b_{n}}^{k}=\emptyset \\ \Omega_{1}^{k}=\Omega_{10}^{k}\cup \Omega_{11}^{k}\cup \ldots \cup \Omega_{1b_{n}}^{k}, & {\;\Omega }_{10}^{k}\cap \Omega_{11}^{k}\cap \ldots \cap \Omega_{1b_{n}}^{k}=\emptyset . \end{array}$$

Corresponding to Equation (11), we let the subpartition satisfy Equation (19) according to the probability of detection and the probability of false alarm:(20)$$\{\begin{array}{c} P_{d10k}=P_{d11k}=\ldots =P_{d1b_{n}k}=\frac{\;P_{dk}}{b_{n}}\;\\ P_{f10k}=P_{f11k}=\ldots =P_{f1b_{n}k}=\frac{P_{fk}}{b_{n}}\; \end{array}$$

The possibility transition matrix is as follows: $$PT_{k}=\;(\;\begin{array}{ccccc} {\left(1-P_{ek}\right)}^{b_{n}+1} & \left(\begin{array}{c} nb_{n}+1\\ 1 \end{array}\right){\left(1-P_{ek}\right)}^{b_{n}}P_{ek} & \ldots & \left(\begin{array}{c} nb_{n}+1\\ b_{n} \end{array}\right){\left(P_{ek}\right)}^{b_{n}}(1-P_{ek}) & {\left(P_{ek}\right)}^{b_{n}+1}\\ \left(\begin{array}{c} nb_{n}+1\\ 1 \end{array}\right){\left(1-P_{ek}\right)}^{b_{n}}P_{ek} & {\left(1-P_{ek}\right)}^{b_{n}+1} & \cdots & \vdots & \vdots \\ \vdots & \vdots & \ddots & \vdots & \vdots \\ {\left(P_{ek}\right)}^{b_{n}+1} & \left(\begin{array}{c} nb_{n}+1\\ b_{n} \end{array}\right){\left(P_{ek}\right)}^{b_{n}}(1-P_{ek}) & \cdots & \left(\begin{array}{c} nb_{n}+1\\ 1 \end{array}\right){\left(1-P_{ek}\right)}^{b_{n}}P_{ek} & {\left(1-P_{ek}\right)}^{b_{n}+1} \end{array})$$

## 4. Hard Decision Fusion 

For the *m*th cluster, we can get its binary decision *u_cm_*∈{0, 1} according to Equation (16) and its corresponding probability of false alarm *P_fcm_* and the probability of detection *P_dcm_*. For M clusters, ***U*** = [*u_c_*_1_, *u_c_*_2_, …, *u_cM_*] is the decision vector of all CHs. Then, the FC combines all binary decisions from all clusters into a final decision. The likelihood ratio test at the FC is: (21)$$L\left(U\right)=\frac{P\left\{U|H_{1}\right\}}{P\left\{U|H_{0}\right\}}=\frac{P\left\{u_{c_{1}},\;u_{c_{2}},\ldots ,u_{c_{M}}|H_{1}\right\}}{P\left\{u_{c_{1}},\;u_{c_{2}},\ldots ,u_{c_{M}}|H_{0}\right\}}\begin{array}{c} \begin{array}{c} H_{1}\\ > \end{array}\\ \begin{array}{c} <\\ H_{0} \end{array} \end{array}T_{0}$$
where *T*_0_ is determined using the global desired probability of false alarm at the FC.

Here, the optimal decision rule of the FC can be derived as [2]:(22)$$\overset{M}{\underset{m=1}{\sum }}w_{m}u_{c_{m}}\begin{array}{c} \begin{array}{c} H_{1}\\ > \end{array}\\ \begin{array}{c} <\\ H_{0} \end{array} \end{array}T_{0}$$
and the coefficients {*w_m_*} are determined using the probability of false alarm *P_fcm_* and the probability of detection *P_dcm_*:$$w_{m}=\{\begin{array}{ll} \log\frac{P_{dc_{m}}}{P_{fc_{m}}} & if\;\;u_{c_{m}}=1,\;\;\;\;\;m=1,\;2,\ldots ,\;M\\ \log\frac{1-P_{dc_{m}}}{1-P_{fc_{m}}} & if\;\;u_{c_{m}}=0,\;\;\;\;\;m=1,\;2,\ldots ,\;M \end{array}$$

## 5. Simulation and Results

In this section, we present the performances of the clustered distributed system with the proposed SH-DFS using Monte Carlo simulation results. We assumed that local sensors with random initial energy from 0.1 to 0.5 J/battery were randomly deployed in an area with dimensions 100 m × 100 m. The FC was at (50 m, 50 m). In addition, we made the *k*th local sensor’s probability of false alarm *P_fk_* ~ U(0.0, 0.2). The probability of error bit *P_ek_* in the non-ideal channel was assumed to be 0.05, and it was the same for each local sensor for convenience.

For the *k*th local sensor, its observation *y_k_* followed a Gaussian distribution:(23)$$\{\begin{array}{l} f\left(y_{k}|H_{1}\right)=\frac{1}{\sqrt{2\pi }\sigma }\exp\left\{-\frac{{\left(y_{k}-E\left(S+N\right)\right)}^{2}}{2\sigma^{2}}\right\}\\ f\left(y_{k}|H_{0}\right)=\frac{1}{\sqrt{2\pi }\sigma }\exp\left\{-\frac{{\left(y_{k}-E\left(N\right)\right)}^{2}}{2\sigma^{2}}\right\} \end{array}$$
where *E*(*N*) is the mean value of noise, *E*(*S* + *N*) is the mean value of signal and noise, and *σ*^2^ is the variance of the Gaussian noise.

### 5.1. Energy Consumption

Figure 7 illustrates the energy consumption of the distributed systems with the conventional hard decision, SH-DFS, the conventional soft decision, and the centralized system. In this paper, the conventional hard decision was that each local sensor made a one-bit hard decision according to its observation and transmitted it to the FC under the non-ideal channel; then, the FC combined all decisions from local sensors into a global decision. In addition, the conventional soft decision in this paper was that each local sensor made a multi-bit decision according to its observation and transmitted it to the FC under the non-ideal channel; then, the FC combined all decisions from local sensors into a global decision. Figure 7 shows the comparisons of four different systems by using the number of alive nodes with the round increasing, in which Con/Hard represents the distributed system with the conventional hard decision, SH-DFS represents the distributed system with the proposed method in this paper, Con/Soft represents the distributed system with conventional soft decision, and the Centralized represents the centralized system.

We can find that the centralized system consumes the most energy because raw information was transmitted by all local sensors to the FC, although it has the optimal detection performance because there was little loss of information. Conversely, the system with a conventional hard decision consumed the least energy because the least information (a one-bit decision) was transmitted by all local sensors to the FC. The energy consumption in the proposed method mainly included the energy consumption for clustering in every round and the energy consumption for bits transmission. However, the conventional soft decision fusion mainly included energy consumption for bits transmission. On the surface, the proposed method consumed extra energy for clustering. However, Figure 7 shows that the system with conventional soft decision fusion (a three-bit decision) consumed more energy than the system with the proposed method (a three-bit decision in the soft decision, M = 4). This is reasonable for two reasons. The first reason is that, in the system with the conventional soft decision fusion, every local sensor needed to transfer a soft decision to the FC. However, in the clustering network system with the proposed method, every local sensor only needed to transfer a soft decision to its corresponding CH, which had a shorter distance. In addition, those CHs only needed to transfer one bit to the FC in the proposed method. These help the system with the proposed method consume less energy for bits transmission. The second reason is that the cluster was reconfigured using FLS and the fuzzy c-means clustering algorithm in every round, which helped the system share the overload in all local sensors. Furthermore, it made the number of alive nodes in the proposed method be more than that in the conventional soft decision fusion method in every round. 

### 5.2. Different SNRs

Performances were analyzed by using the probability of detection versus SNR. Here, let the desired global probability of false alarm at the FC be 0.01.

Figure 8 shows the comparison of the performances of the distributed systems with the conventional hard decision, SH-DFS, the conventional soft decision, and of the centralized system. Here, let the number of local sensors be 30. We found that the centralized system had the best performance because there was little loss of information in it. The system with SH-DFS (a two-bit decision, M = 4) significantly outperformed the system which only used a hard decision fusion (Con/Hard). In addition, although the system with the SH-DFS had similar performance with the system that only used soft decision fusion (a two-bit decision), it was better than the system with Con/Soft because it consumed less energy according to Figure 7.

Figure 9 shows the comparison of the performances of the clustered distributed systems with SH-DFS (M = 4, *b* = 2, *P_e_* = 0.05), but with a different number of local sensors. The number of local sensors was fixed at 5, 15, 25, and 35. From Figure 9, we found that the number of local sensors had a smaller influence at a low SNR than that at a high SNR. Furthermore, the probability of detection increased with the number of local sensors increasing.

Figure 10 shows the comparison of the performances of the clustered distributed systems with SH-DFS (N = 30, *b* = 2, *P_e_* = 0.05), but with a different number of clusters. We observed that with the rise of the cluster number, the probability of detection increased.

Figure 11 shows the comparison of the performances of the clustered distributed systems with SH-DFS (N = 30, M = 4, *P_e_* = 0.05), where *b* is fixed at 2, 3, and 4. With *b* increasing, which means more information was transmitted, the performance of the system with SH-DFS also increased. Compared with the centralized system, we found that the performance of the system with SH-DFS was close to the performance of the centralized system when *b* was 3 or 4. Therefore, two-bit soft fusion or three-bit soft fusion can be applied to improve the performance of the Con/Hard, where only a one-bit decision is transmitted. At the same time, when we need to get a tradeoff between communication costs and detection performance, it is meaningless to let *b* (the number of bits in b-bit soft fusion) be more than 4.

Figure 12 shows the comparison of the performances of the clustered distributed systems with SH-DFS (N = 30, M = 4, *b* = 2), where *P_e_* = 0 and *P_e_* = 0.05 respectively. It was clear that the non-ideal communication channel had a negative influence on the clustered distributed system. It was necessary to take the non-deal communication channel into consideration.

### 5.3. Different Probabilities of the Desired Global False Alarm at the FC

Performances were analyzed by using the probability of detection versus the global desired probability of false alarm at the FC. Here, let SNR be 10 dB. 

Figure 13 shows the comparison of the performances of the distributed systems with the conventional hard decision, SH-DFS, the conventional soft decision, and of the centralized system. Here, let the number of local sensors be 30. We found that the centralized system had the optimal performance. The system with SH-DFS (a two-bit decision, M = 4) significantly outperformed the system which only used hard decision fusion (Con/Hard). In addition, the system with the SH-DFS had a similar performance with the system that only used soft decision fusion (a two-bit decision).

Figure 14 shows the comparison of the performances of the clustered distributed systems with SH-DFS (M = 4, *b* = 2, *P_e_* = 0.05), but with a different number of local sensors. The number of local sensors was fixed at 5, 15, 25, and 35. It was clear that given the same SNR and the global desired probability of false alarm at the FC, the probability of detection increased along with the increase in the number of local sensors. Furthermore, it was also clear that given the same SNR and the number of local sensors, the probability of detection increased along with the increase of the global desired probability of false alarm at the FC. 

Figure 15 shows the comparison of the performances of the clustered distributed systems with SH-DFS (N = 30, *b* = 2, *P_e_* = 0.05), but with a different number of clusters. With the rise of the cluster number, the probability of detection was increased. However, for the system with 30 local sensors, when the cluster number changed from 2 to 4, its probability of detection had a more significant improvement than that when the cluster number changed from 4 to 8.

Figure 16 shows the comparison of the performances of the clustered distributed systems with SH-DFS (N = 30, M = 4, *P_e_* = 0.05), where *b* was fixed at 2, 3, and 4. When *b* increased, the performance of the system with SH-DFS also increased because more information was transmitted and fused to make a global final decision. In addition, we found that the performance of the system with SH-DFS was close to the performance of the centralized system with the increase of *b*. The system could achieve a probability of detection of about 0.8 when the probability of false alarm was fixed at 0.1.

Figure 17 shows the comparison of the performances of the clustered distributed systems with SH-DFS (N = 30, M = 4, *b* = 2), where *P_e_* = 0 and *P_e_* = 0.05. It was clear that the non-ideal communication channel had a negative influence on the clustered distributed system.

### 5.4. Soft Decision Fusion with Equal Gain Combining (EGC)

Soft decision fusion with equal gain combining (SH-EGC) is a soft fusion rule, which has a good detection performance and is easy to apply [29]. It has been studied and widely used in cooperative spectrum sensing systems. Here, we compare our method SH-DFS with SH-EGC. Let N be 30, M be 4, *P_e_* be 0.05, and SNR be 2dB for all local sensors.

Figure 18 shows the comparison of the detection performances of SH-DFS and SH-EGC. We found that SH-DFS (*b* = 3, 4) had a good performance as well as SH-EGC without quantization, and the little difference between them could be ignored. Although the largest number of quantization bits was decided by the system in practice, the two-bit quantization or the three-bit quantization used in the proposed method could usually be accepted, which helps the system achieve a probability of detection of about 0.8 and 0.9 when the probability of false alarm was fixed at 0.1. 

## 6. Conclusions

In this paper, we propose a soft–hard combination decision fusion scheme for the clustered distributed detection system with multiple sensors and non-ideal communication channels. Simulation results show that the performance of the clustered distributed system with the proposed SH-DFS significantly outperforms the system with only a conventional hard decision fusion. At the same time, the clustered distributed system with the proposed SH-DFS had a similar performance to the centralized data fusion system when a three-bit decision or a four-bit decision was made in SH-DFS, but it consumed less energy than the centralized system. From the simulation results, we found that it was meaningless to let *b* (the number of bits in soft decision fusion in SH-DFS) be more than 4 when we want to get a tradeoff between communication costs and detection performance. In addition, by clustering the distributed system in every round using FLS and the fuzzy c-means clustering algorithm, the lifetime of the system could be extended because the load could be shared in all local sensors. The proposed method could be applied to the cooperative spectrum sensing system, to the underwater target detection system and the radar target detection system and so on. 

## Figures and Tables

**Figure 1 sensors-18-04370-f001:**
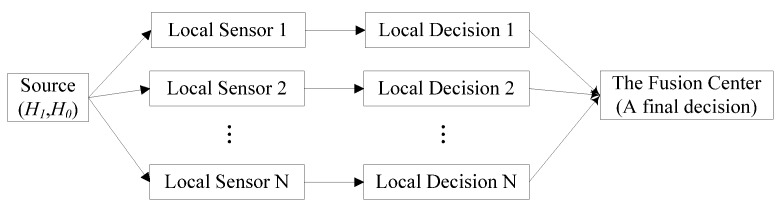
A typical distributed detection system with multiple sensors.

**Figure 2 sensors-18-04370-f002:**
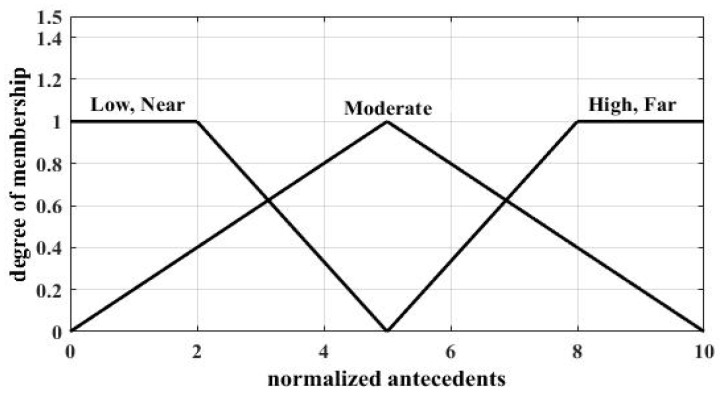
MFs for normalized antecedents.

**Figure 3 sensors-18-04370-f003:**
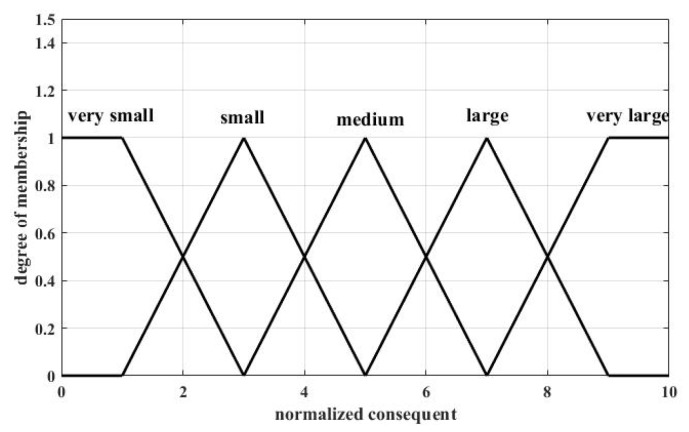
MFs for normalized consequents.

**Figure 4 sensors-18-04370-f004:**
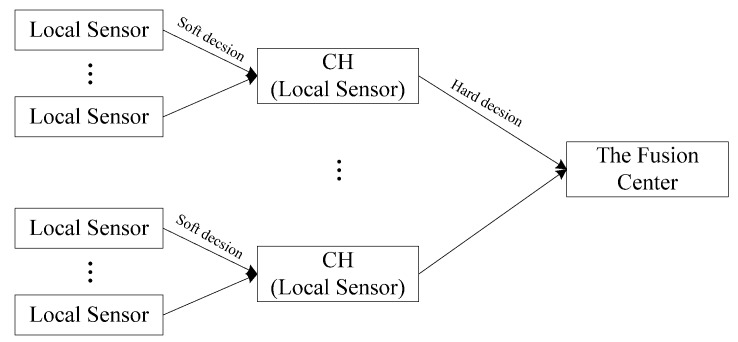
The proposed clustered system.

**Figure 5 sensors-18-04370-f005:**
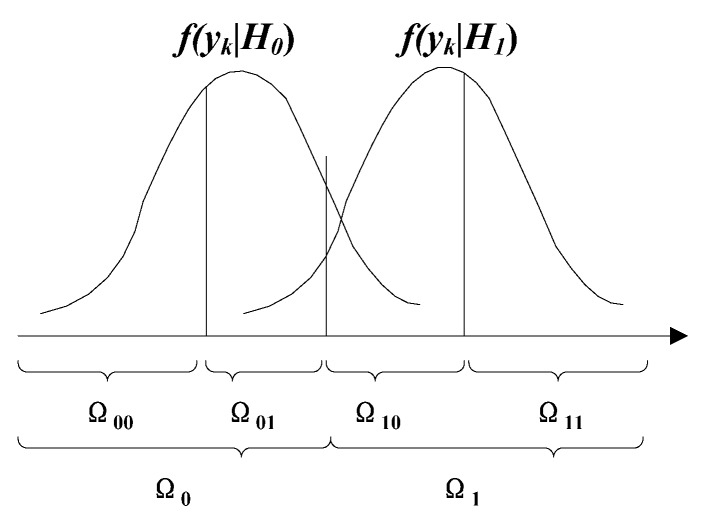
The subpartitioning figure in the 2-bit decision case.

**Figure 6 sensors-18-04370-f006:**
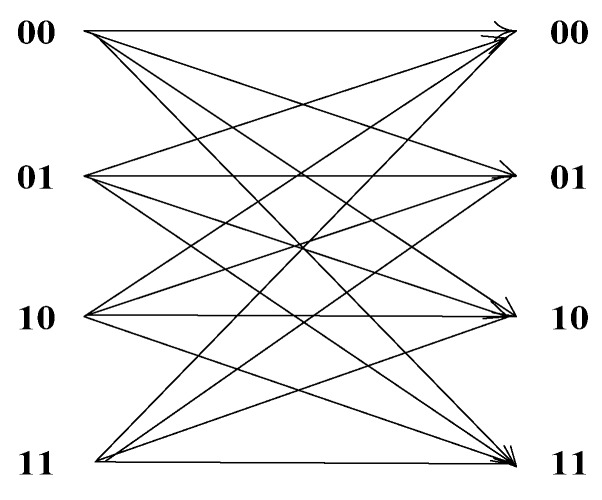
The station transition diagram in the 2-bit decision case.

**Figure 7 sensors-18-04370-f007:**
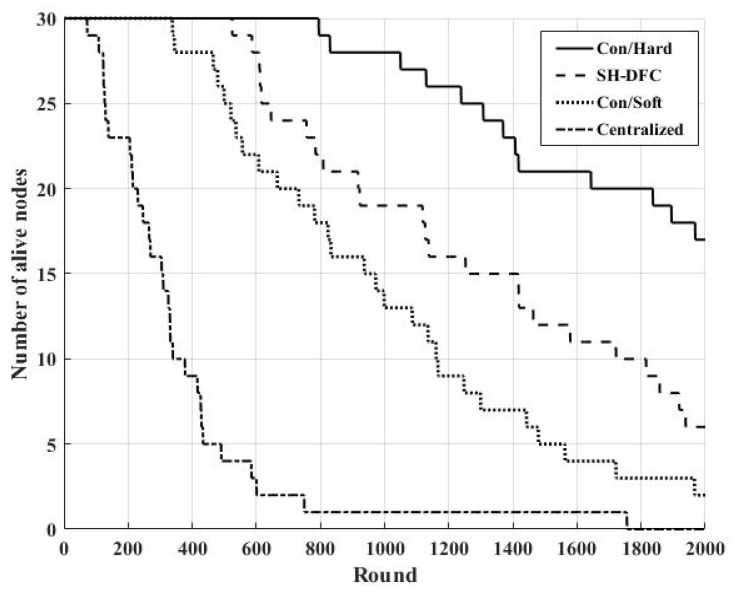
The number of alive nodes versus the round.

**Figure 8 sensors-18-04370-f008:**
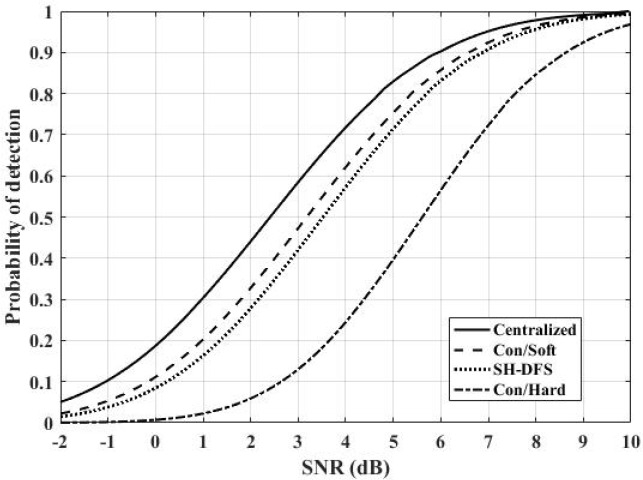
The probability of detection of four different methods versus SNR.

**Figure 9 sensors-18-04370-f009:**
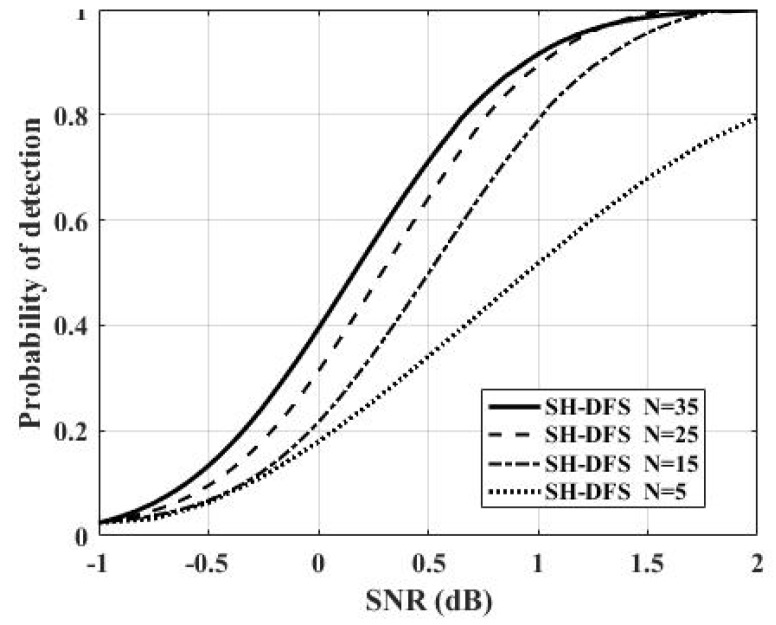
The probability of detection of the clustered distributed system with SH-DFS at a different number of local sensors versus SNR.

**Figure 10 sensors-18-04370-f010:**
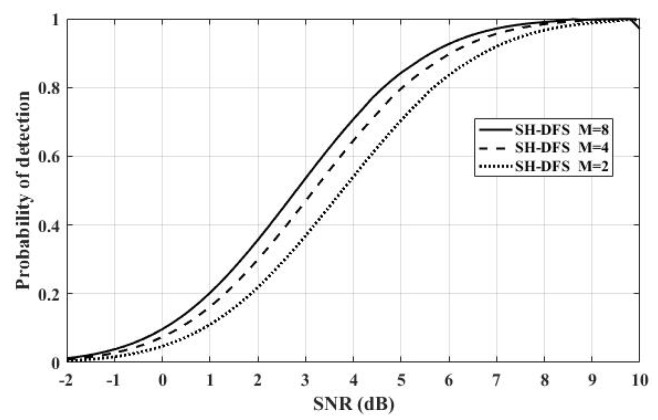
The probability of detection of the clustered distributed system with SH-DFS and different clusters versus SNR.

**Figure 11 sensors-18-04370-f011:**
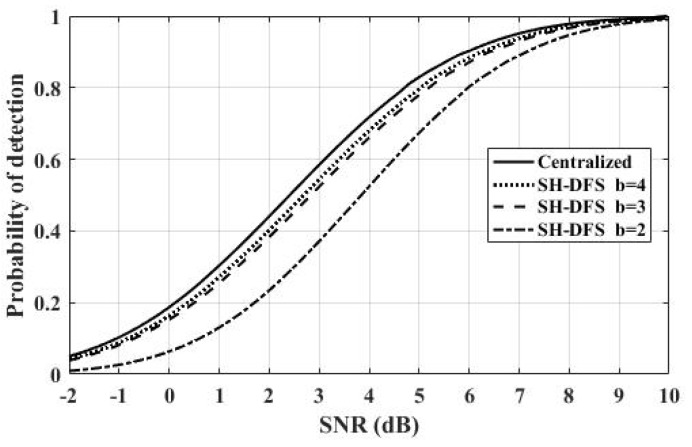
The probability of detection of the clustered distributed system with SH-DFS (*b* = 2, 3, 4) and the centralized system versus SNR.

**Figure 12 sensors-18-04370-f012:**
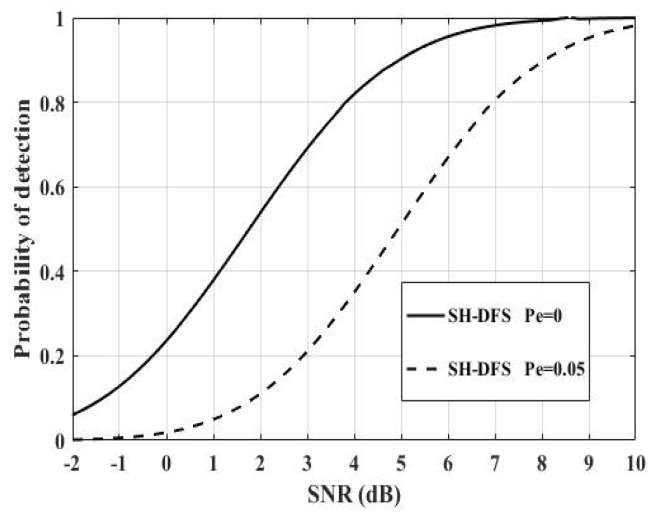
The probability of detection of the clustered distributed system with SH-DFS, where *P_e_* = 0 and *P_e_* = 0.05.

**Figure 13 sensors-18-04370-f013:**
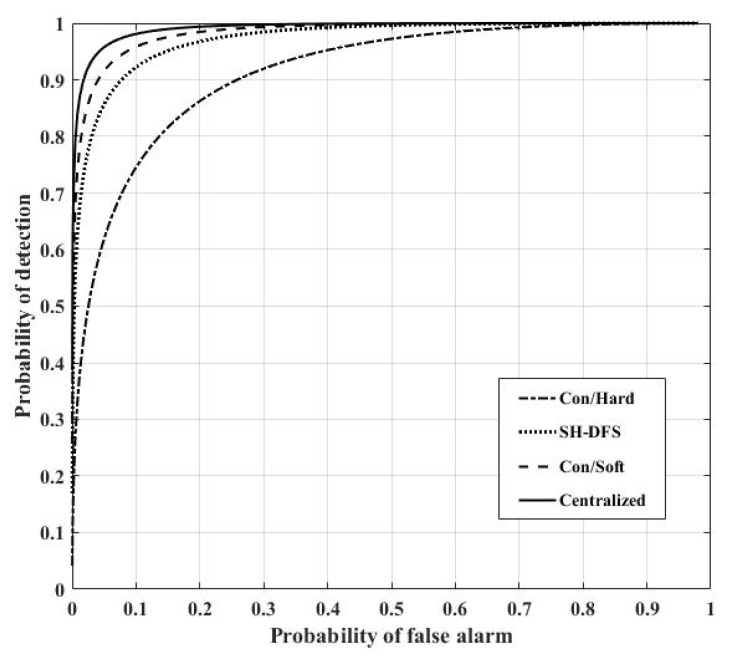
The probability of detection of four different methods versus the global desired probability of false alarm at the FC.

**Figure 14 sensors-18-04370-f014:**
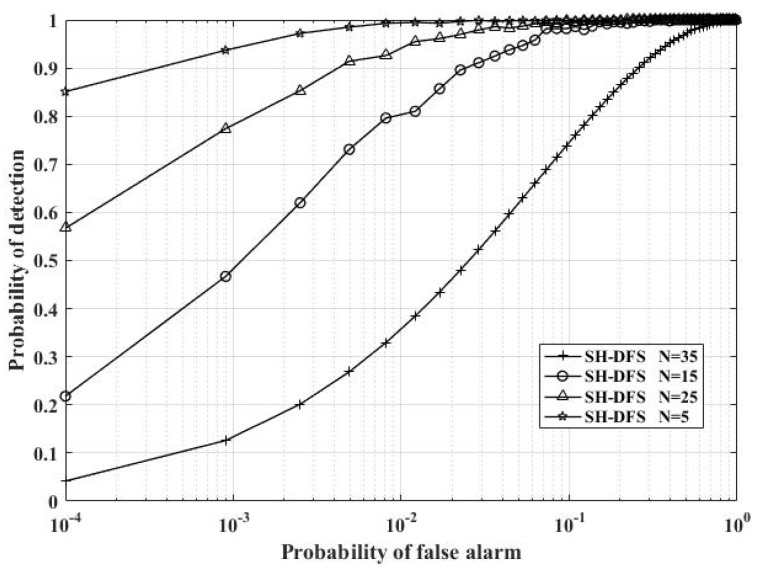
The probability of detection of the clustered distributed system with SH-DFS at a different number of local sensors versus the global desired probability of false alarm at the FC.

**Figure 15 sensors-18-04370-f015:**
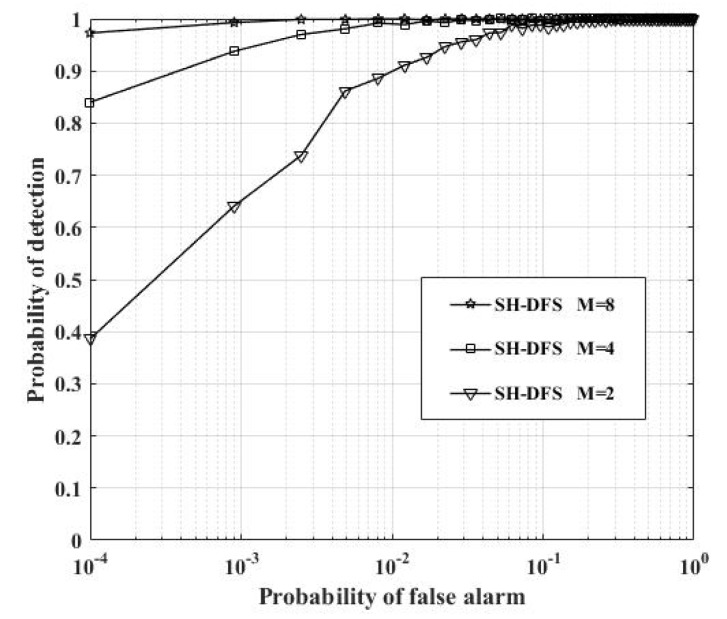
The probability of detection of the clustered distributed system with SH-DFS at a different cluster number versus the global desired probability of false alarm at the FC.

**Figure 16 sensors-18-04370-f016:**
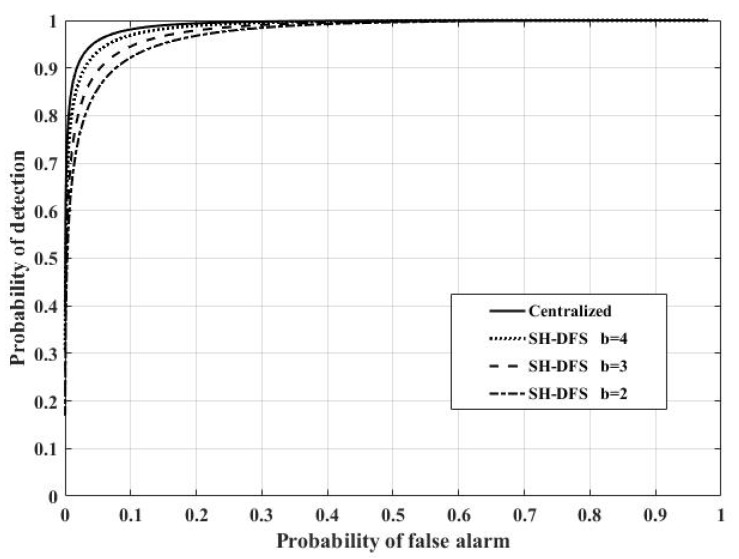
The probability of detection of the clustered distributed system with SH-DFS (*b* = 2, 3, 4) and the centralized system versus the global desired probability of false alarm at the FC.

**Figure 17 sensors-18-04370-f017:**
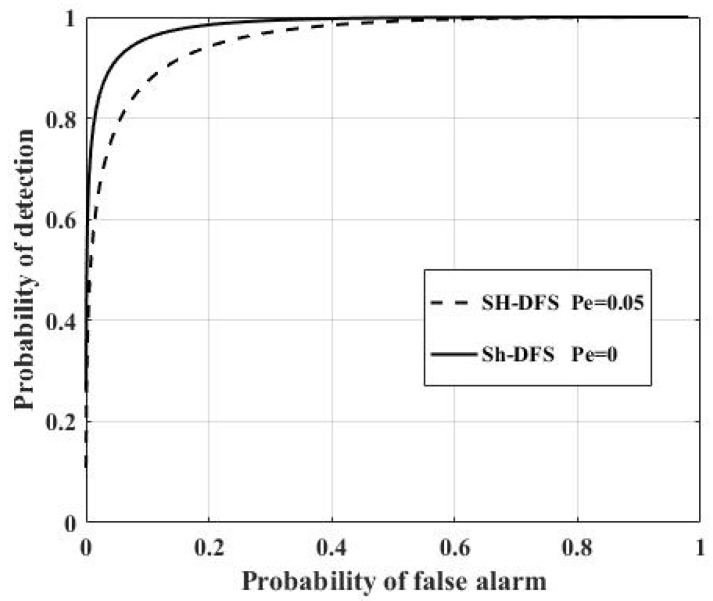
The probability of detection of the clustered distributed system with SH-DFS with *P_e_* = 0 and *P_e_* = 0.05 versus the global desired probability of false alarm at the FC.

**Figure 18 sensors-18-04370-f018:**
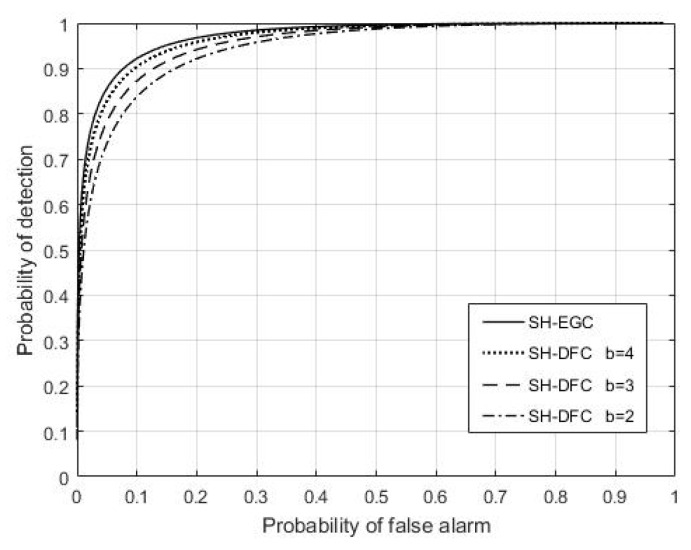
The probability of detection of two methods versus the global desired probability of false alarm at the FC.

**Table 1 sensors-18-04370-t001:** Rules in FLS.

Rule	Antecedent 1	Antecedent 2	Consequent
1	low	near	medium
2	low	moderate	small
3	low	far	very small
4	moderate	near	large
5	moderate	moderate	medium
6	moderate	far	small
7	high	near	very large
8	high	moderate	large
9	high	far	medium

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
