# Peer review of "A Soft–Hard Combination Decision Fusion Scheme for a Clustered Distributed Detection System with Multiple Sensors"

_sensors, 2018, doi:10.3390/s18124370_

Reviewer 1 Report
In the abstract expand CH. Typically, the very first time abbreviation is used, it should be in expanded form. Follow the same rule all through the paper.
On line 81, expand the abbreviation CHEF.
Line 110: remove "remainder of this"
Title of Fig. 1 could be "A typical distributed detection system with multiple sensors."
Line 118: replace "including" by "consisting of"
Line 126: What is RS? expand RS
Line 151: replace "local" by "local sensor"
Address all the grammatical errors in the paper
Define the quantities: i) SNR, ii) Communication cost used in the paper. Also, explain how thresholds are calculated.
Some comparisons of the simulation results of the paper with existing theoretical results in the literature is needed.
Some applications for the work presented in the paper should be included.
Conclusions could be made better using a couple of numerical examples.
Author Response
We would like to thank the reviewer for the careful and thorough reading of this manuscript and for the thoughtful comments and constructive suggestions, which help to improve the quality of this manuscript. Our changes in the paper follow the reviewer’s comments in red.
Point 1: In the abstract expand CH. Typically, the very first time abbreviation is used, it should be in expanded form. Follow the same rule all through the paper.
Response 1: We have expanded CH used in the first time.
(line 19) In clusters, each local sensor transmits its local multiple-bit soft decision to its corresponding cluster head (CH)under the non-ideal channel, in which a simple and efficient soft decision fusion method is used.
Point 2: On line 81, expand the abbreviation CHEF.
Response 2: We have expanded CHEF on line 81 used in the first time.
(line 87-88) In the cluster head election mechanism using fuzzy logic (CHEF), CHs are selected based on two parameters which are proximity distance and energy [23].
Point 3: Line 110: remove "remainder of this"
Response 3: We have removed "remainder of this" on line 110.
(line 120) The paper is organized as follows.
Point 4: Title of Fig. 1 could be "A typical distributed detection system with multiple sensors."
Response 4: We have changed the title of Fig 1.
(line 126) Figure 1. A typical distributed detection system with multiple sensors.
Point 5: Line 118: replace "including" by "consisting of"
Response 5: We have replaced "including" by "consisting of".
(line 129) We consider a distributed detection system consisting of N local sensors.
Point 6: Line 126: What is RS? expand RS
Response 6: We have replaced "an RS" by "a local sensor".
(line 138) Therefore, two antecedents of a local sensor are considered in our designed FLS:
• Antecedent 1: every local sensor’s remaining energy.
• Antecedent 2: every local sensor’s distance to the FC.
Point 7: Line 151: replace "local" by "local sensor"
Response 7: We have replaced "local" by "local sensor".
(line 163) For every input (x1, x2) of each local sensor, the output is computed by
Point 8: Line 118: Define the quantities: i) SNR, ii) Communication cost used in the paper. Also, explain how thresholds are calculated.
Response 8: Detailed description in the attachment
Point 9: Some comparisons of the simulation results of the paper with existing theoretical results in the literature is needed.
Response 9: We have added some comparisons and the corresponding reference.
Detailed description in the attachment.
Point 10: Some applications for the work presented in the paper should be included.
Response 10: We have given some simple applications in Section 5 for the work presented in the paper.
(line 434-436) In addition, by clustering the distributed system in every round using FLS and the fuzzy c-means clustering algorithm, the lifetime of the system could be extended because the load can be shared in all local sensors. The proposed method could be applied to the cooperative spectrum sensing system, to the underwater target detection system and even the radar target detection system and so on.
Point 11: Conclusions could be made better using a couple of numerical examples.
Response 11: We have added some numerical values to explain our conclusion.
(line 407-408) The system can achieve a probability of detection of about 0.8 when the probability of false alarm is fixed at 0.1.
Point 12: Address all the grammatical errors in the paper.
Response 12: We have corrected those grammatical errors we could find in the paper.

Reviewer 2 Report
Most references are old. The original point is the hypothesis of a non-ideal transmission channel. The paper is clearly presented.
Some remarks
The clustering technique is clearly explained, but I have not understood at which level the clusters computation is done, in the fusion center or locally and distributed at each node. The first version needs energy to configure the network at each round.
Likewise at the level of the transmission, as an error is bit inversion, soft decision coding must have a role to play, and this point is not discussed in the paper.
Text editing
line 207 link omega and indices
line 220 sensor
Author Response
We would like to thank the reviewer for the careful and thorough reading of this manuscript and for the thoughtful comments and constructive suggestions, which help to improve the quality of this manuscript. Our changes in the paper follow the reviewer’s comments in red.
Point 1: Most references are old.
Response 1: We have added some new references.
(line 71-73)However, most of those proposed methods are computationally complex especially with multilevel quantization and didn’t take the non-ideal communication channel into consideration. In [25], the author optimized the number of reporting bits to maximize the network’s throughput in quantized cooperative spectrum sensing. In [26], the number of reporting bits and the combining weight were jointly optimized to maximize the probability of detection.
(line 89-92)In the cluster head election mechanism using fuzzy logic (CHEF), CHs are selected based on two parameters which are proximity distance and energy [23]. In [27], the energy efficient structured clustering algorithm (EESCA) is proposed, in which CH is elected based on average communication distance and lingering energy. In [28], the author proposed the Adaptive Dynamic Clustering (ADC) to minimize the cluster head and improve the network’s routing problem.
(line 500-510)
25. Nguyen-Thanh, N.; Ciblat, P.; Maleki, S.; Nguyen, V.T. How Many Bits Should Be Reported in Quantized Cooperative Spectrum Sensing? IEEE Wirel. Commun. Lett. 2015, 4, 465–468, doi:10.1109/LWC.2015.2437879.
26. Abdi, Y.; Ristaniemi, T. Joint local quantization and linear cooperation in spectrum sensing for cognitive radio networks. IEEE Trans. Signal Process. 2014, 62, 4349–4362, doi:10.1109/LWC.2015.2437879.
27. Padmanaban, Y.; Muthukumarasamy, M. Energy-efficient clustering algorithm for structured wireless sensor networks. IET Networks 2018, 7, 265–272, doi:10.1049/iet-net.2017.0112.
28. Sivasakthiselvan, S.; Nagarajan, V. Mobility management and adaptive dynamic clustering for mobile wireless sensor networks. Proc. 2017 IEEE Int. Conf. Commun. Signal Process. ICCSP 2017 2018, 2018-January, 2246–2251, doi:10.1109/ICCSP.2017.8286816.
Point 2: The clustering technique is clearly explained, but I have not understood at which level the clusters computation is done, in the fusion center or locally and distributed at each node. The first version needs energy to configure the network at each round.
Response 2: The clusters computation is done in the fusion center, and in our proposed method, the network is reconfigured at each round. In section 5(Simulation and Results), Fig 7 shows the network’s energy consumption by illustrating the number of alive local sensors with the round increasing, in which the energy consumption for clustering in the proposed method(SH-DFC) is included. We add some explanations and analysis in this section.
(line 311-325)We can find that the centralized system consumes the most energy because raw information is transmitted by all local sensors to the FC, although it has the optimal detection performance because there is small loss of information. Conversely, the system with conventional hard decision consumes the least energy because the least information (a one- bit decision) is transmitted by all local sensors to the FC. The energy consumption in the proposed method mainly includes the energy consumption for clustering in every round and the energy consumption for bits transmission. However, the conventional soft decision fusion mainly includes energy consumption for bits transmission. On the surface, the proposed method consumes extra energy for clustering. But Figure 7 shows that the system with conventional soft decision fusion (a 3-bit decision) consumes more energy than the system with the proposed method(a 3-bit decision in the soft decision, M=4). It is reasonable because two reasons. The first reason is that, in the system with the conventional soft decision fusion, every local sensor needs to transfer soft decision to the FC. But in the clustering network system with the proposed method, every local sensor only needs to transfer soft decision to its corresponding CH, which has a shorter distance. In addition, those CHs only need to transfer one bit to the FC in the proposed method. These help the system with the proposed method consumes less energy for bits transmission. The second reason is that the cluster is reconfigured by FLS and the fuzzy c-means clustering algorithm in every round, which helps the system share the overload in all local sensors. And it makes the number of alive nodes in the proposed method be more than that in the conventional soft decision fusion method in every round.
Point 3: Likewise at the level of the transmission, as an error is bit inversion, soft decision coding must have a role to play, and this point is not discussed in the paper.
Response 3: We add the discussion about the soft decision coding. Detailed description in the attachment.
Point 4: Text editing
line 207 link omega and indices
line 220 sensor
Response 4: We corrected these errors.
Detailed description in the attachment.
